

# Genome-wide association study reveals genetic basis and candidate genes for chlorophyll content of leaves in maize (*Zea mays* L.)

Peng Liu[1,2], Chenchaoyang Xiang[2], Kai Liu[2,3], Hong Yu[2,4], Zhengqiao Liao[1], Yaou Shen[2], Lei Liu[1] and Langlang Ma[2]

[1] Mianyang Teachers College, Mianyang, Sichuan, China
[2] Sichuan Agricultural University, Chengdu, Sichuan, China
[3] Leshan Academy of Agricultural Sciences, Leshan, Sichuan, China
[4] Zigong Academy of Agricultural Sciences, Zigong, Sichuan, China

Corresponding authors
Lei Liu, 33020897@qq.com
Langlang Ma, sxyljxml@163.com

## ABSTRACT

The chlorophyll content (CC) directly affects photosynthesis, growth, and yield. However, the genetic basis of CC is still unclear in maize (*Zea mays* L.). Here, we conducted a genome-wide association study using mixed linear model for CC of the fifth leaves at seedling stage (CCFSS) and the ear leaves at filling stage (CCEFS) for 334 maize inbred lines. The heritability estimates for CCFSS and CCEFS, obtained *via* variance components analysis using the lme4 package in R, were 70.84% and 78.99%, respectively, indicating that the CC of leaves is primarily controlled by genetic factors. A total of 15 CC-related SNPs and 177 candidate genes were identified with a *p-value* < $4.49 \times 10^{-5}$, which explained 4.98–7.59% of the phenotypic variation. Lines with more favorable gene variants showed higher CC. Meanwhile, Gene Ontology (GO) analysis implied that these candidate genes were probably related to chlorophyll biosynthesis. In addition, gene-based association analyses revealed that six variants in *GRMZM2G037152*, *GRMZM5G816561*, *GRMZM2G324462*, and *GRMZM2G064657* genes were significantly (*p-value* < 0.01) correlated with CC, of which *GRMZM2G064657* (encodes a phosphate transporter protein) and *GRMZM5G816561* (encodes a cytochrome P450 protein) were specifically highly expressed in leaves tissues. Interestingly, these candidate genes were previously reported to involve in the regulation of the contents of chlorophyll in plants or Chlamydomonas. These results may contribute to the understanding of genetic basis and molecular mechanisms of maize CC and the selection of maize varieties with improved CC.

## INTRODUCTION

Chlorophyll, being the primary photosynthetic pigment, plays a crucial role in capturing energy, mainly in the blue and red wavelengths, and facilitating electron transport within the chloroplasts of higher plants. An increase in chlorophyll content significantly boosts the plant's ability to absorb light, thereby enhancing the efficiency of photosynthesis

(*Li et al., 2023*). In the case of Chinese cabbage, a stably inherited etiolated mutation resulted in a notable decrease in chlorophyll content, which led to a reduction in photosynthetic capacity and retarded chloroplast development, as compared to the wild type (*Li et al., 2019*). This underscores the intimate relationship between plant chlorophyll content and both leaf photosynthetic efficiency and crop yield. Similar observations have been presented across different crop species. For instance, in wheat, a strong correlation ($R^2 = 0.8$) has been established between chlorophyll potential and grain yield (*Sid'ko et al., 2017*). In maize, *Yan et al. (2021)* conducted a field experiment demonstrating that high-yield maize cultivars were depended on several key factors, including elevated photosynthetic capacity, extended photosynthetic duration, an optimal leaf area index (LAI), and a higher chlorophyll content coupled with a lower chlorophyll a/b ratio. Furthermore, as an indicator of plant chlorophyll content, the mean Soil Plant Analysis Development (SPAD) value is significantly correlated with multiple yield-determinative traits such as number of kernels per row, number of kernel rows per ear, ear weight, grain yield, and *etc*., (*Ghimire, Timsina & Nepal, 2015*). These findings highlight the importance of optimizing chlorophyll content in crops to enhance their photosynthetic performance and ultimately improve yields.

Chlorophyll is located within the thylakoid membrane of the chloroplast, consisting of chlorophyll a and chlorophyll b. The chlorophyll biosynthesis pathway in higher plants is complex, consisting of at least 15 steps, from L-glutamyl-tRNA to chlorophyllide b (*Beale, 2005*). So far, more than 17 enzymes were identified involving the chlorophyll biosynthesis in plants, such as L-Glutamyl-tRNA (GluTR), magnesium chelatase I subunit (MgCh), Chlorophyll synthase (CHLG), aminolaevulinic dehydratase (ALAD), Chlorophyllide a oxygenase (CAO), Coproporphyrinogen oxidase (CPO), uroporphyrinogen III decarboxylase (UROD), and numerous others (*Tripathy & Pattanayak, 2012*; *Zhao et al., 2020*). Furthermore, *Zhang et al. (2021)* reported that the GUN4:bilin adducts likely regulate chlorophyll biosynthesis by delivering protoporphyrin to CHLH subunit of Mg chelatase. *Geng et al. (2023)* reported that the knockdown of a chloroplast-localized gene *PCD8* would lead to chloroplast damage and caused a necrotic phenotype in Arabidopsis. Maize (*Zea mays* L.), as a globally cultivated crop that serves as a staple food, animal feed, and industrial raw material. Understanding the genetic basis of chlorophyll content will be helpful to improve maize photosynthetic efficiency and yield. By combining bulked segregant analysis and complementation allelic test, researchers identified the key gene, *ZmCRD1*, encoding magnesium-protoporphyrin IX monomethyl ester cyclase (MgPEC), which affects chlorophyll content in a chlorophyll-deficient maize mutant and its wild-type (*Xue et al., 2022*). By combining a semidominant mutant allele of *oy1* and a cis-regulatory modifier named *very oil yellow1* (*vey1*), the chlorophyll content was changed between different maize inbred lines (*Khangura, Johal & Dilkes, 2020*). Despite these advancements, the comprehensive molecular mechanisms underlying chlorophyll synthesis and metabolism in maize, as well as the candidate genes related to these processes, remain elusive, which needs further research to fully unravel their intricacies. Hence, conducting an in-depth analysis of the genetic basis of maize leaf CC is imperative

for cultivating novel maize varieties that exhibit enhanced photosynthetic efficiency, thereby boosting overall maize productivity.

Owing to the comprehensive scanning of the genome and exploiting of numerous ancient recombination events and linkage disequilibrium (LD), genome-wide association study (GWAS) has become a powerful tool to elucidate the genetic basis of complex quantitative traits in plants (*Wang et al., 2023*; *Susmitha et al., 2023*). Recent advances in next-generation sequencing for GWAS have enabled high-resolution single nucleotide polymorphism (SNP) discovery, revolutionizing genetic dissection of complex crop traits and accelerating research progress in crop improvement. So far, numerous candidate genes and quantitative trait nucleotides (QTNs) associated with chlorophyll contents (CC) in crops have been identified by GWAS. For instance, *Wang et al. (2015)* performed a GWAS using 529 rice (*Oryza sativa* L.) accessions and identified 46 significant CC-related loci. Around these loci, they identified a major causal gene *grain number, plant height, and heading date7* (*GHD7*), which decreased chlorophyll content by downregulating the expression of genes involved in the biosynthesis of chlorophyll and chloroplast development. *Dhanapal et al. (2016)* conducted a GWAS in soybean, identifying 52 unique SNPs and 155 genes that were related to chlorophyll content. In maize, based on an association panel consisting of 290 maize inbred lines, 10 co-located QTNs and 69 candidate genes were detected to be associated with CC (*Xiong et al., 2023*). *Jin et al. (2023)* conducted a GWAS of maize chlorophyll traits based on 378 maize inbred lines with extensive natural variation and found 19 SNPs containing 76 candidate genes related to leaf senescence, photosynthesis, and plant developmental processes. Nevertheless, the highest phenotypic variance explained (PVE) by these CC-related genetic loci is less than 10%, indicating that numerous genetic factors remain to be discovered and understood.

In this study, we measured the chlorophyll content in maize and performed a GWAS to identify the variants and candidate genes influencing chlorophyll content. Subsequently, we analyzed the genetic structure of chlorophyll, aiming to provide a theoretical basis for breeding optimal-photosynthetic efficiency of maize lines.

## MATERIALS AND METHODS

### Plant materials

The association panel utilized in this study included 334 maize lines provided by the Maize Institute of Sichuan Agricultural University. These lines were collected from the breeding program of Southwest China and consist of tropical, non-stiff stalk (NSS), stiff stalk (SS), and other unique germplasms (*Zhang et al., 2016*; Table S1). The majority of these accessions belong to the mid to late maturity group, with maturity periods ranging from 100 to 125 days. The panel was rigorously evaluated across three distinct environments with significant differences in climatic conditions, namely Chongzhou (CZ, Sichuan Province; 30.30° N, 103.07° E) and Ya'an (YA, Sichuan Province; 29.59° N, 102.57° E) in 2021, as well as Xishuangbanna (XSBN, Yunnan Province; 22.0° N, 100.79° E) in 2022. The evaluation employed a completely randomized design with three replicates for consistency and reliability. Each line was grown in a single row with row length of 3 m and row

distance of 0.7 m. A standard corn management practices were applied during the cultivation.

## Phenotypic data collection and analysis

The SPAD values were collected from five plants with consistent growth condition of each line to symbolize the chlorophyll contents using SPAD 502 Plus Chlorophyll Meter (a handheld SPAD instrument). At the seedling stage (30 days after sowing) and the grain filling stage (5 days after pollination), the CC was measured at the middle parts of the fifth leaf and the ear leaf, respectively. Each plant was measured three times, after which the mean value was recorded as the leaf CC. The descriptive statistics of CCFSS and CCEFS in each environment, including mean, maximum (Max), minimum (Min), standard deviation (SD), coefficient of variation (CV), skewness, and kurtosis of CC were analyzed using the psych package (*Revelle, 2024*) in R 4.4.2 (*R Core Team, 2024*). To evaluate multi-environment experimental data, the BLUP values were computed using a liner mixed model for the estimation of random effects with lme4 R package. The ANOVA analysis was performed to calculate the variance components of each trait, including genotypes (G), environments (E), and interactions between genotype and environment (G × E). The broad-sense heritability ($H^2$) was estimated using the following formula (*Knapp, 1986*):

$$H^2 = \frac{\sigma_g^2}{\sigma_g^2 + \frac{\sigma_{ge}^2}{e} + \frac{\sigma_e^2}{er}}$$

Here, $\sigma_g^2$, $\sigma_e^2$, and $\sigma_{ge}^2$ represent genetic variance, residual error variance, and the variance of genetic × environmental interaction, respectively. The letters "e" and "r" denote the number of environments and the number of independent replicates, respectively.

## GWAS

The genotype of the association panel was genotyped using the Maize SNP50 BeadChip, which consisted of 56,110 SNPs (*Zhang et al., 2016*). Those SNPs with minor allele frequency (MAF) < 0.05, missing rate > 20%, or heterozygosity > 20% were considered as low-quality variations, which had been filtered out. Then, a total of 43,728 high-quality SNPs were used for GWAS based on MLM model in the GEMMA package. Meanwhile, population structure (Q = 6) reported in the previous study was used as a covariate in the GWAS model (*Zhang et al., 2016*). The SNPs were considered as trait-associated QTNs with the *p-value* less than $1/N = 4.49 \times 10^{-5}$ (N = 22,277). The N denotes the effective marker number of independent tests, which was calculated using simpleM function in R 4.4.2 package. The quantile–quantile (Q-Q) and Manhattan plots for GWAS were generated by using the CMplot function in R 4.4.2 package (https://github.com/YinLiLin/CMplot). The PVE of each significant associated SNP was calculated according to the formula as follows (*Liu et al., 2024*):

$$PVE = \frac{2\hat{\beta}^2 MAF(1 - MAF)}{2\hat{\beta}^2 MAF(1 - MAF) + \left(se\left(\hat{\beta}\right)\right)^2 2N * MAF(1 - MAF)}$$

where $\hat{\beta}$ is the effect estimate of genetic variants, MAF is the minor allele frequency of genetic variants, N is the sample size, and $se\left(\hat{\beta}\right)$ is the standard error of the effect. The raw phenotype and genotype data for maize lines are available in the figshare database with the accession link of https://doi.org/10.6084/m9.figshare.26355523.v1.

## Analysis of candidate genes

According to the LD decay of this panel, gene models located within the 220 kb flanking regions of all trait-associated QTNs were identified as potential trait-related candidate genes. The function descriptions, gene ontology (GO) terms, and Kyoto Encyclopedia of Genes and Genomes (KEGG) pathways of these genes were annotated based on the Annotation database of maize B73 RefGen v2 in MaizeGDB (https://www.maizegdb.org/, accessed 25 May 2024). Enrichment analysis of GO and KEGG were performed using an online tool OmicShare (https://www.omicshare.com/).

## Gene-based association analysis

The variations within the gene bodies and 2,000 bp upstream regions of candidate genes were obtained by DNA re-sequencing from 77 maize lines (*Liu et al., 2024*). The MLM model was tested to detect key variations associated with CC of maize at the *p-value* of 0.05. The LD decay between pairwise SNPs was calculated using LDBlockShow software (version 1.4) (*Dong et al., 2021*).

## Gene expression patterns

The expression levels of candidate genes of distinct tissues at different development stages in maize were obtained from a previous study (*Stelpflug et al., 2016*). The heatmap was drawn using the pheatmap function in R 4.4.2 package (*Kolde, 2019*; *R Core Team, 2024*).

## Statistical analysis

The two-sided *t*-test of the CC-related traits between two types of haplotypes was performed in Excel 2021. Box plots were created using the R ggplot2 package (https://ggplot2.tidyverse.org/).

## RESULTS

### Phenotypic variation

Phenotypic data of 334 maize inbred lines were collected across three experiment environments (Chongzhou, CZ; Xishuangbanna, XSBN; and Ya'an, YA;). The values of mean, Max, Min, SD, CV, skewness, and kurtosis of CC showed significant variability (Table 1 and Fig. 1). The SPAD values of CCFSS varied between 29.78–58.54, 28.89–53.56, and 31.53–55.15 across CZ, XSBN, and YA environments, respectively, with the CVs of 10.10%, 10.25%, and 11.62%. While, the range SPAD values of CCEFS in CZ, XSBN, and YA were 42.70–66.09, 34.67–68.40, and 38.55–64.84 respectively, with the CVs of 7.72%,

**Table 1 Phenotypic variations of two traits in 334 maize inbred lines.**

| Trait | Environment | Mean | SD | Min | Max | CV | Skew | Kurtosis | $H^2$ |
|-------|-------------|------|------|-------|-------|-------|-------|----------|-------|
| CCFSS | CZ | 43.98 | 4.44 | 29.78 | 58.54 | 10.10 | 0.07 | 0.17 | 70.84% |
| | XSBN | 38.55 | 3.95 | 28.89 | 53.56 | 10.25 | 0.52 | 0.48 | |
| | YA | 42.14 | 4.90 | 31.53 | 55.15 | 11.62 | 0.02 | −0.40 | |
| | BLUP | 41.54 | 2.55 | 34.93 | 50.39 | 6.14 | 0.32 | 0.23 | |
| CCEFS | CZ | 54.34 | 4.20 | 42.70 | 66.09 | 7.72 | −0.18 | −0.06 | 78.99% |
| | XSBN | 53.18 | 3.96 | 34.67 | 68.40 | 7.45 | −0.42 | 1.76 | |
| | YA | 53.63 | 4.17 | 38.55 | 64.84 | 7.78 | −0.39 | 0.31 | |
| | BLUP | 53.69 | 2.83 | 45.58 | 62.13 | 5.28 | −0.25 | 0.08 | |

**Note:**
CCFSS, chlorophyll contents of fifth leaves in seedling stage; CCEFS, chlorophyll contents of ear leaves in filling stage; SD, standard deviation for the population; Min, minimum value; Max, maximum value; CV, coefficient of variation; $H^2$, broad-sense heritability.

7.45%, and 7.78%. In addition, the BLUP values of CCFSS and CCEFS were calculated to eliminate the environmental deviation, with an average of 41.54 (ranging from 34.93 to 50.39) and 53.69 (ranging from 45.58 to 62.13), separately. Moreover, the absolute values of skewness and kurtosis for CCFSS and CCEFS across all environments and BLUP were less than 1.0 except for kurtosis of CCEFS in XSBN, indicating that CC followed normal distributions, and was controlled by numerous genes. Besides, the SPAD values shown significant positive correlations between any two environments or BLUP, indicating that there is a certain connection between the chlorophyll content of different development stages (Fig. S1). The values of broad-sense heritability ($H^2$) for CCFSS and CCEFS were 70.84% and 78.99%, respectively, which confirmed that the CC of leaves were mainly controlled by genetic factors (Table 1).

## QTNs associated with CC by GWAS

We used a MLM method with a threshold $p$-$value$ of $4.49 \times 10^{-5}$ to identify CC-related genetic loci. In total, 15 CC-related SNPs were detected. Among them, 11 SNPs were associated with CCFSS, and four were associated with CCEFS (Figs. 2, 3 and Table S2). The PVE values for these SNPs were 4.98–7.59%, indicating that chlorophyll content in maize was controlled by multiple mini-effect genetic loci. Notably, four SNPs were identified as co-located loci, which were detected in at least two different environments (including BLUP). These common loci indicate that they have more stable genetic effects, which should be attentioned in further studies. Three of the co-located SNPs (PZE-101214133, PZE-106069023, and PZsE-108003930) linked to CCFSS were suited in chromosomes 1, 6, and 8, respectively. One SNP (SYN23593), associated with CCEFS, was located in chromosome 7. For PZE-101214133, PZE-106069023, and PZE-108003930, the mean value of CC of all germplasms with the minor allele was significantly ($p$-$value$ < 0.01) higher than that of germplasms with the major allele across the population (Table 2). However, the opposite performance was observed in SYN23593, where the average CC value of germplasms with the major allele was significantly higher than that of germplasms with the minor allele. (Table 2). Specifically, for CCFSS, the most significantly associated

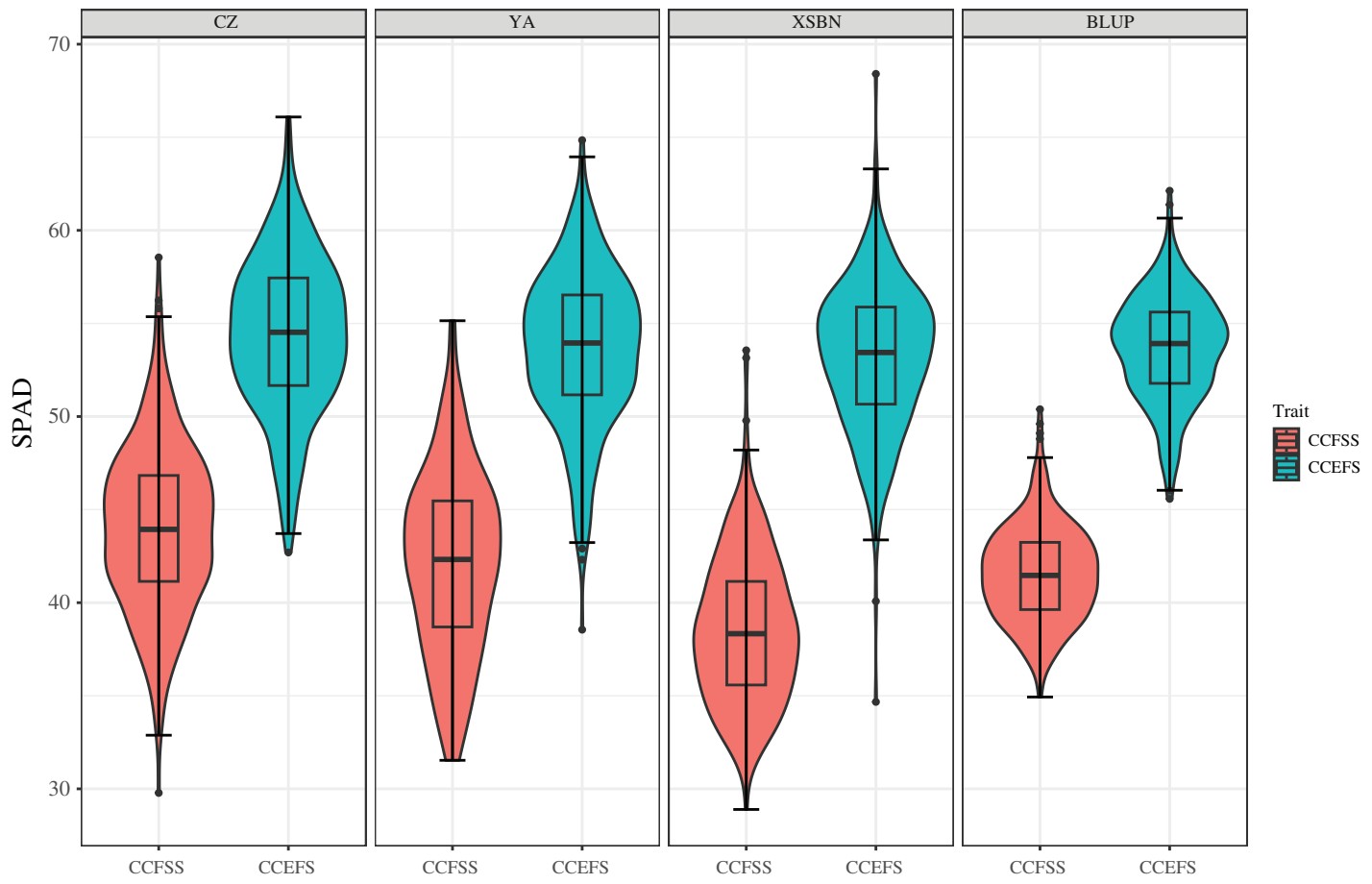

**Figure 1 Phenotypic variations of CCFSS and CCEFS in different environments.** CZ, YA, XSBN, and BLUP represent Chongzhou, Ya'an, Xishuangbanna, and the best linear unbiased prediction, respectively. The X-axis and Y-axis represent different traits and the chlorophyll contents (SPAD), respectively. CCFSS and CCEFS represent chlorophyll contents of fifth leaves at seedling stage and ear leaves at filling stage, respectively.

marker was PZE-101222689 ($p\text{-}value = 4.68 \times 10^{-7}$), located on chromosome 1 and explaining 7.37% of the phenotypic variation (Table S2). This SNP was located in the first intron of the gene *GRMZM2G416388*, which encodes a cystathionine beta-synthase (CBS) family protein. CBS proteins were known as energy sensors that regulate protein activities *via* their adenosyl ligand binding capacity. Specifically, in Arabidopsis, CBSX2 has been shown to inhibit the activities of m-type thioredoxins (TRXs) toward two chloroplast TRX-related targets, thus regulating plant growth (*Baudry et al., 2022*). For CCEFS, the most significantly associated marker was SYN23593 ($p\text{-}value = 2.95 \times 10^{-7}$), with a PVE of 7.59 (Table S2). This marker was located in the 3′-UTR of the gene *GRMZM2G057296*, which encodes a pectin lyase-like superfamily protein (PEL). In rice, the *PEL* gene takes part in the regulation of plant growth and leaf senescence (*Leng et al., 2017*).

We then analyzed the superior allele ratios in 30 maize lines widely utilized in the Southwest of China to evaluate the superior allele application of the 15 significant SNPs during maize breeding. Since higher CC is important for crop production, we considered the allele with the positive effect as the superior allele. Conversely, the alleles associated

with lower SPAD values were designated as the inferior alleles. The superior allele percentage of all CC-related SNPs ranged from 0% (PZE-102047226, PZE-103162301, and PZE-104118967) to 70% (PZE-101222689) (Fig. 4). Notably, only three SNPs (PZE-101222689, PZE-106067897, and PUT-163a-149111517-965) had a superior allele ratio greater than 50%. However, eight SNPs (PZE-101068676, PZE-101214118, PZE-101214133, PZE-102047226, PZE-103162301, PZE-104118967, PZE-107016552, and PZE-108003930) possessed superior allele ratios of <10%, implying that these eight SNPs should be given priority consideration in molecular marker-assisted breeding to modify the CC of maize. Especially, two of the four co-located SNPs (PZE-101214133 and PZE-108003930) respectively existed in three and two maize lines, which should be paid more attentions. Among the 30 elite inbred lines, the maize line Mian723 owned the greatest number of superior alleles, possessing higher CCFSS and CCEFS of 47.79 and 62.13 in BLUP, respectively. The line F06 had only one superior allele, with the lower CCFSS and CCEFS of 38.03 and 49.98, respectively. Therefore, Mian723 and F06 had great utility values to display higher CC by providing or integrating more superior alleles.

## Analysis of candidate genes

To further select the potential CC-related candidate genes, we searched the genes within 220-kb flanking regions of the 15 significant genetic loci identified by GWAS. In total, 177 candidate genes were detected, of which 131 genes had functional annotations (Table S3). Among these genes, 16 common genes were situated in the LD regions of SNPs PZE-101214118 and PZE-101214133, which were close to each other (<2,000 bp). Similarly, two genes simultaneously located in the LD region of PUT-163a-149111517-965 and PZE-106069023. According to the annotations, *GRMZM5G820904* encodes a translocon at the outer envelope membrane of chloroplasts 75-III protein (TOC 75-III). As previously reported, TOC gene initiated the import process of thousands of nuclear precursor proteins, which are crucial for chloroplast formation, plant growth and development (*Richardson et al., 2014*). Another gene *GRMZM2G164084* encodes an RNA polymerase sigma factor, which is essential to life and controls the process of transcription (*Borukhov & Nudler, 2008*). While the RNA polymerase sigma factor controls all transcription initiation steps and the stimulation of the primary steps in RNA synthesis (*Vishwakarma & Brodolin, 2020*). In Arabidopsis, gene SIG2 takes part in the transcription of several chloroplast tRNA genes possibly couples translation and pigment synthesis in chloroplasts (*Kanamaru & Tanaka, 2004*). The E1-E2 ATPase encoded by *GRMZM2G324462* is also known as *P*-type ATPase. In Arabidopsis, a *P*-type ATPase involved in regulating the expression of a downstream gene ALA10, impacts the fatty acyl composition of chloroplast phosphatidylcholine, changing chlorophyll contents (*Botella et al., 2016*). *GRMZM5G883222* encodes a phosphatidylinositol-4-phosphate 5-kinase family protein, which is involved in the initiation of chloroplast division by fusing a part of sequence of the prokaryotic FtsZ (a prokaryotic homolog of tubulin) (*Shimada et al., 2004*). In addition, the four co-located SNPs, PZE-101214133, PZE-106069023, PZE-108003930, and SYN23593 harbored 16, 9, 11, and 6 genes, respectively. Notably, the phosphate transporter encoded by *GRMZM2G064657* (PZE-106069023), which was confirmed to

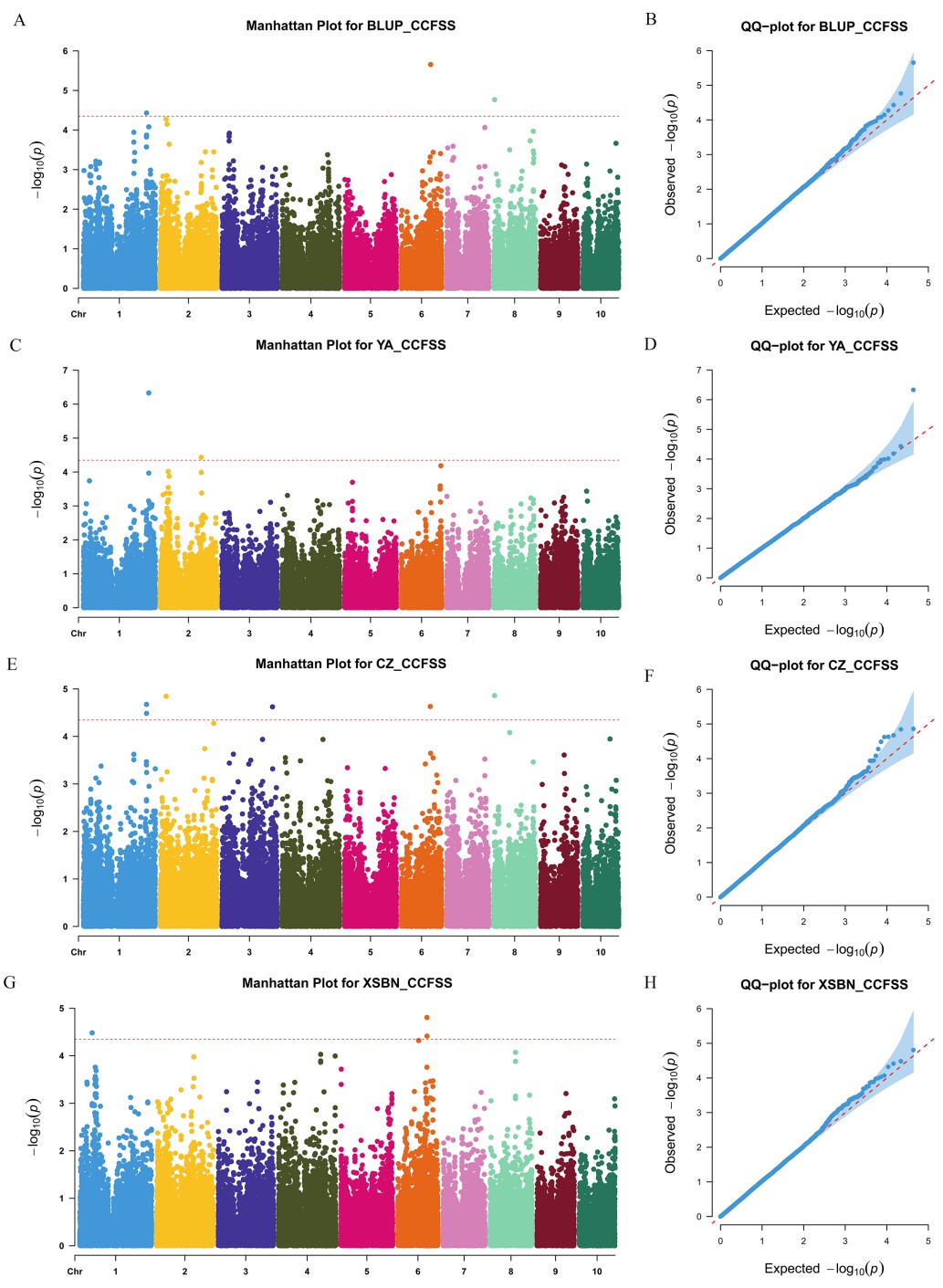

**Figure 2 Manhattan and Q-Q plots of a genome-wide association analysis of CCFSS in different environments.** Manhattan and Q-Q plots for BLUP (A and B), YA (Ya'an, Sichuan) (C and D), CZ (Chongzhou, Sichuan) (E and F), and XSBN (Xishuangbanna, Yunnan) (G and H), respectively. X-axis represents chromosomal positions. Y-axis represents -$\log_{10}$ (*p-values*) of each marker. The dotted lines indicate the genome-wide significance threshold (*p-value* = $4.49 \times 10^{-5}$).

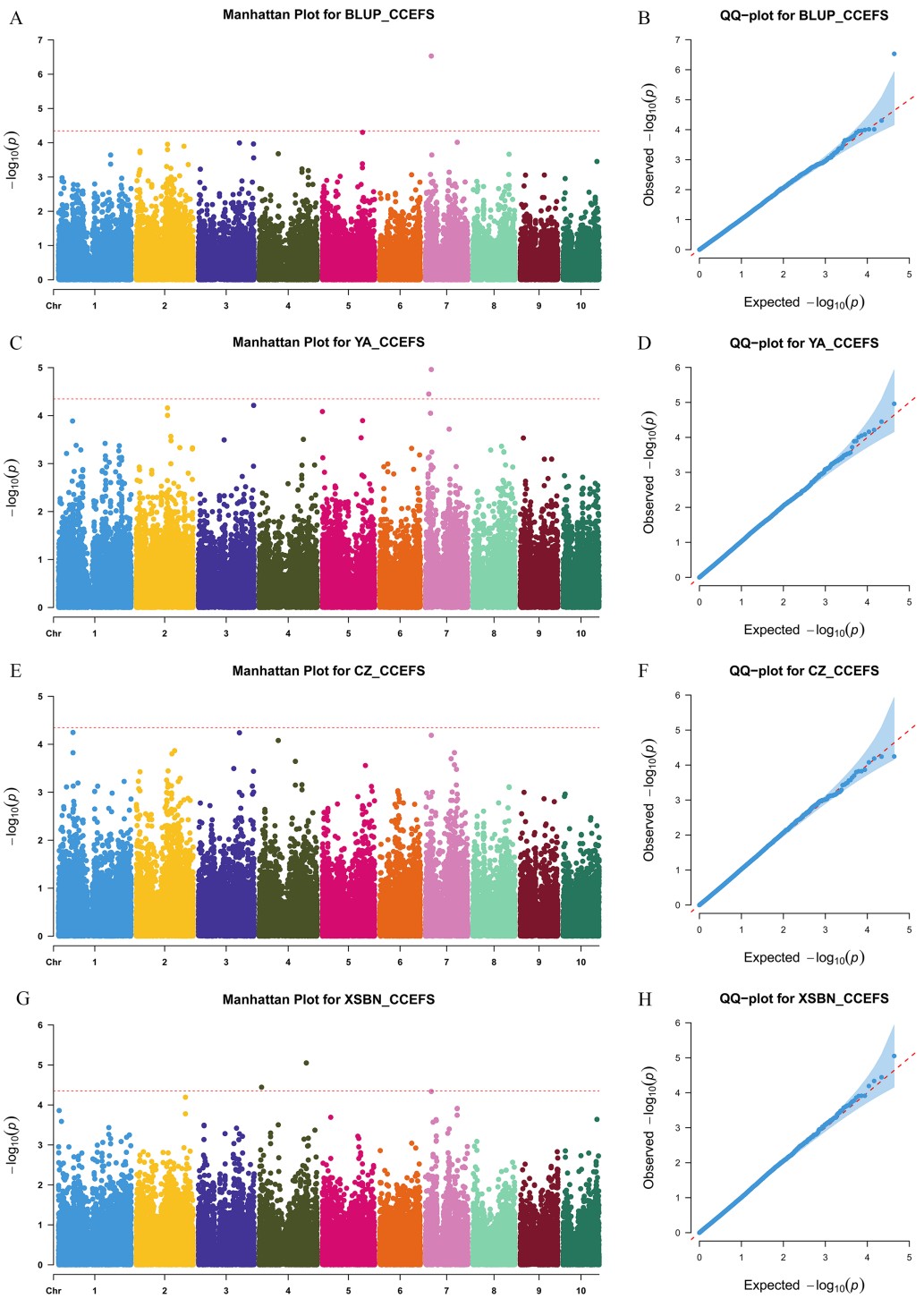

**Figure 3 Manhattan and Q-Q plots of a genome-wide association analysis of CCEFS in different environments.** Manhattan and Q-Q plots for BLUP (A and B), YA (Ya'an, Sichuan) (C and D), CZ (Chongzhou, Sichuan) (E and F), and XSBN (Xishuangbanna, Yunnan) (G and H), respectively. X-axis represents chromosomal positions. Y-axis represents -$\log_{10}$ (*p-values*) of each marker. The dotted lines indicate the genome-wide significance threshold (*p-value* = $4.49 \times 10^{-5}$).

**Table 2 T-*test* of two alleles of each co-located CC-related SNPs in different environments.**

| Trait | Loci | Chrom | Position | Env | Allele | No. | Phenotype | *P-value* |
|-------|------|-------|----------|-----|--------|-----|-----------|-----------|
| CCFSS | PZE-101214133 | 1 | 264522231 | CZ | A | 74 | 45.425 | 2.96E−04 |
| | | | | | G | 233 | 43.345 | |
| | | | | BN | A | 65 | 39.716 | 2.44E−03 |
| | | | | | G | 188 | 38.038 | |
| | | | | YA | A | 66 | 43.706 | 3.97E−03 |
| | | | | | G | 217 | 41.749 | |
| | | | | BLUP | A | 75 | 42.462 | 1.13E−04 |
| | | | | | G | 235 | 41.189 | |
| | PZE-106069023 | 6 | 122398586 | CZ | C | 126 | 45.152 | 1.10E−04 |
| | | | | | A | 199 | 43.21 | |
| | | | | BN | C | 106 | 40.056 | 2.33E−07 |
| | | | | | A | 162 | 37.563 | |
| | | | | YA | C | 116 | 43.399 | 5.98E−04 |
| | | | | | A | 183 | 41.424 | |
| | | | | BLUP | C | 127 | 42.429 | 3.19E−07 |
| | | | | | A | 202 | 40.98 | |
| | PZE-108003930 | 8 | 3922598 | CZ | A | 53 | 45.678 | 1.34E−03 |
| | | | | | G | 252 | 43.516 | |
| | | | | BN | A | 42 | 39.975 | 6.95E−03 |
| | | | | | G | 211 | 38.172 | |
| | | | | YA | A | 48 | 43.251 | 8.75E−02 |
| | | | | | G | 233 | 41.927 | |
| | | | | BLUP | A | 53 | 42.469 | 2.21E−03 |
| | | | | | G | 256 | 41.292 | |
| CCEFS | SYN23593 | 7 | 24409023 | CZ | G | 159 | 53.562 | 1.14E−03 |
| | | | | | A | 164 | 55.076 | |
| | | | | BN | G | 132 | 52.23 | 1.99E−04 |
| | | | | | A | 153 | 53.972 | |
| | | | | YA | G | 149 | 52.809 | 8.85E−04 |
| | | | | | A | 153 | 54.401 | |
| | | | | BLUP | G | 162 | 53.03 | 3.71E−05 |
| | | | | | A | 169 | 54.304 | |

**Note:**
Chr, chromosome; Env, environments; No., number of lines.

affect the chlorophyll contents by regulating the phosphate acquisition in Cucumber (*Naureen et al., 2018*). The pentatricopeptide repeat (PPR) superfamily protein encoded by *GRMZM2G071162* (PZE-106069023) is involved in the post-transcriptional regulation of chloroplast genes, and effects on the biogenesis and functioning of chloroplasts (*Wang et al., 2021*). These results further demonstrate that the candidate genes were potentially associated with CC.

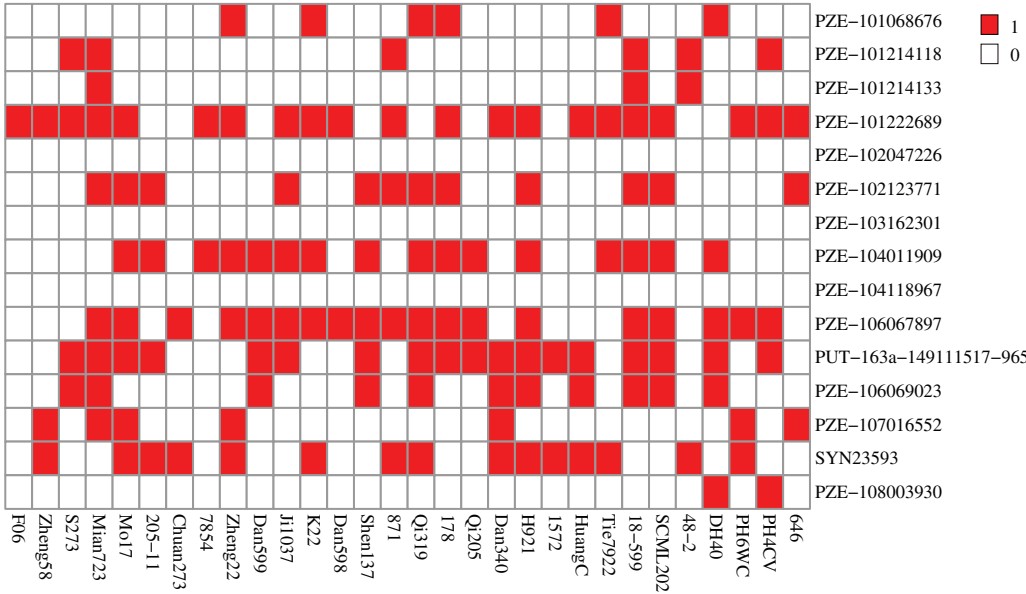

**Figure 4** Heatmap of the distribution of superior alleles in 30 elite maize lines. Red and white colors represent superior and inferior alleles, respectively.

To further reveal the function of these genes, we performed Gene Ontology (GO) and Kyoto Encyclopedia of Genes and Genomes (KEGG) pathway enrichment analyses. As results, GO terms of "peroxidase activity", "oxidoreductase activity", and "inositol phosphate biosynthetic process" and pathways of "phenylpropanoid biosynthesis", "MAPK signaling pathway", and "inositol phosphate metabolism" were significantly enriched ($p$-value < 0.05), indicating that these genes hold significant potential for further research and exploration (Fig. 5, Tables S4, S5). Specially, there are 19 and 22 genes enriched in top 20 GO terms and KEGG pathways respectively, including eight genes that are previously identified as common located genes.

## Gene-based association analysis revealed loci affecting maize CC

To further reveal the key variation loci affecting CC in maize, we conducted gene-based association analyses for the eight hub candidate genes using 77 randomly selected lines from the maize association panel. A total of 678 high-quality variations (542 SNPs, 69 insertions, and 67 deletions) located in the gene regions and their 2,000 bp upstream were obtained by DNA re-sequence (Table S6; *Liu et al., 2024*). Using the MLM model, a total of six variations (four SNPs, one insertion, and one deletion) from four genes (*GRMZM2G037152*, *GRMZM5G816561*, *GRMZM2G324462*, and *GRMZM2G064657*) were significantly ($p$-value < 0.01) associated with maize CC (Table S7). Among them, SNP-1-269340061 was situated in the first exon of gene *GRMZM2G037152* and annotated as a missense variant, which probably alter the corresponding protein sequence. SNP-2-16709401 was annotated as a splice region variant, and located in the first intron on the gene *GRMZM5G816561*. SNP-6-126152668, DEL-6-126298795, SNP-6-126298598, and INS-6-126298833 were all located in the upstream region of the genes *GRMZM2G324462* and *GRMZM2G064657*. *GRMZM2G037152* encodes a GNS1/SUR4 membrane family

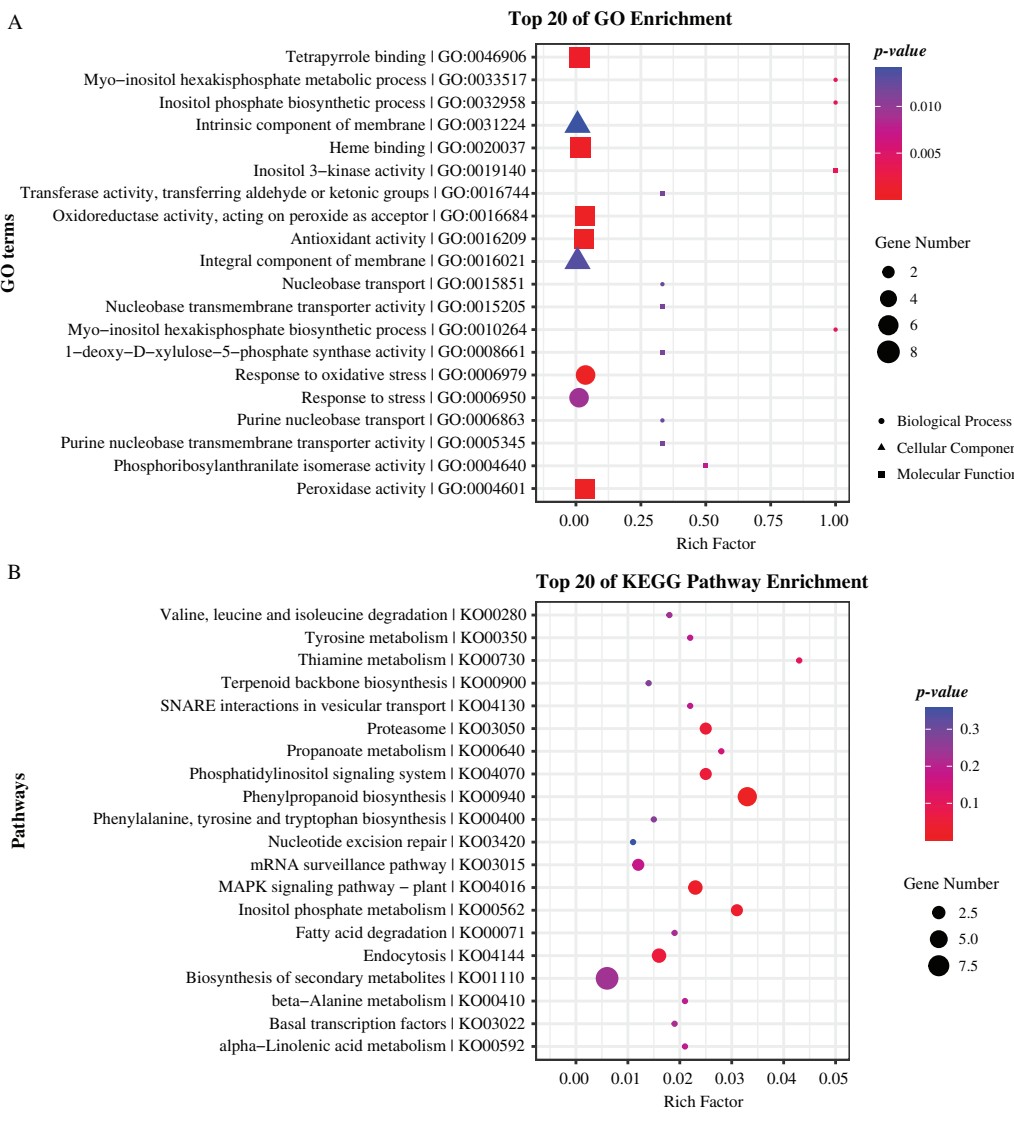

**Figure 5 Enrichment analysis of candidate genes.** (A) GO terms enriched with these specifical genes. (B) KEGG pathways enriched with these specifical genes. Different shapes represent different classes of GO terms. Different sizes represent different number of genes.

protein, which was found to be associated with photosynthetic metabolism and biological processes of photomorphogenesis (*Ma, Ding & Li, 2017*). Cytochrome P450 encoded by *GRMZM5G816561* is one of the most prominent families of oxidoreductases class enzymes. It catalyzes NADPH- and/or O2-mediated hydroxylation reactions in primary and secondary metabolism in various species (*Xu, Wang & Guo, 2015*; *Chakraborty et al., 2023*). *GRMZM2G324462* and *GRMZM2G064657* encodes an ATPase E1-E2 type family protein and a phosphate transporter protein, respectively. As mentioned above, the E1-E2 ATPases also known as P-type ATPases, which is related to the regulation of the contents of chlorophyll in Arabidopsis (*Botella et al., 2016*). The phosphate transporter could regulate phosphate homeostasis and photosynthesis in chlamydomonas (*Tóth et al., 2024*)

and share similar features with chloroplast transit peptides in Arabidopsis (*Versaw & Harrison, 2002*). In particular, the expression level of this gene shown a significant difference between a yellow-green leaf mutant of maize and its wild type (*Li et al., 2021*). Together, these results further demonstrate that the candidate genes were potentially causal genes affecting maize CC.

According to these significant variants, we constructed haplotypes of each gene. For *GRMZM2G037152*, the corresponding SNP (SNP-1-269340061) with allele of A/C divided 77 maize lines into two groups. The mean values of both CCFSS and CCEFS traits in A-type lines were higher than those in C-type maize lines in all environments, except for CCEFS in YA and BLUP. Likewise, two alleles (A/C) of SNP-2-16709401 for *GRMZM5G816561* and two alleles (A/C) of SNP-6-126152668 for *GRMZM2G324462* both divided the 77 lines into two groups. The lines with Hap1 (A) for *GRMZM5G816561* shown higher chlorophyll contents than that with Hap2 (C) in all environments. In contrast, lines with Hap1 (A) for *GRMZM2G324462* shown lower chlorophyll contents than that with Hap2 (C) in all environments. In addition, based on three significant CC-associated variations (DEL-6-126298795, SNP-6-126298598, and INS-6-126298833) from *GRMZM2G064657*, two predominant haplotypes were classified (Hap1: AA-, Hap2: G-G). A t-*test* analysis revealed that the average CCFSS and CCEFS of Hap1 (AA-) lines were significantly ($p$-value < 0.05) higher than those of Hap2 (G-G) lines in all environments, except for CCFSS in YA (Fig. 6). Thus, the Hap1 (AA-) was regarded as a superior haplotype for improving chlorophyll contents in maize.

## Expression patterns of the candidate genes

Based on a public reported gene atlas of maize tissues at different development stages (*Stelpflug et al., 2016*), we examined the expression profiles of all 177 candidate genes and constructed a heatmap (Fig. S2). The expression levels of these candidate genes were varied significantly in different tissues. Furthermore, we focused on eight hub CC-related genes, and found that *GRMZM5G816561* showed a lower expression level in all tissues of different development stages except for some leaf-related tissues. Similarly, *GRMZM2G064657* showed higher expression levels in leaves relative to other tissues. Universally, *GRMZM5G810275*, *GRMZM2G467059*, *GRMZM2G073668*, *GRMZM2G002499*, and *GRMZM2G324462* had a relatively higher level of expression in whole development stages in all tissues (Fig. 7 and Table S8). Gene *GRMZM2G037152* displays a relatively higher expression level in leaf-related and internode-related tissues. These results provide more information for revealing the mechanism of chlorophyll synthesis in further studies.

## DISCUSSION

Chlorophyll is the main pigment which was used to absorb and transform sunlight in plants, and its content directly determines the efficiency of photosynthesis and plant growth. Usually, chlorophyll content of specific plant tissues shows great variation at different developmental stages. For instance, chlorophyll content in wheat flag and second top leaves reached the peak during early grain filling and changed widely among varieties

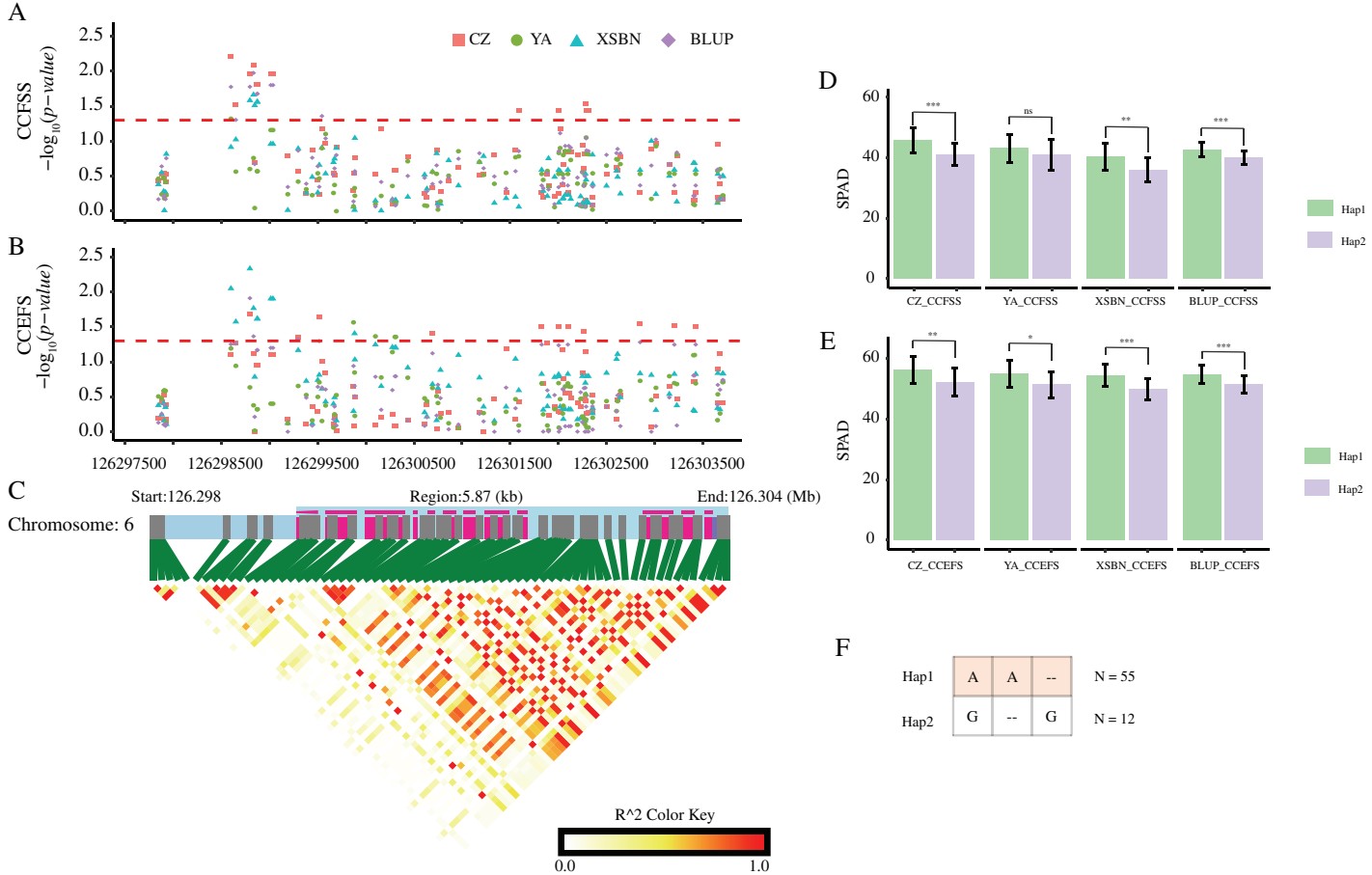

**Figure 6  CC-associated loci in *GRMZM2G064657*.** (A and B) Significant variants associated with CCFSS (A) and CCEFS (B) in the gene body and 2,000 bp upstream of *GRMZM2G064657*. Red, green, blue, and purple scatters represent CZ, YA, XSBN, and BLUP environments, respectively. The y-axis represents negative $\log_{10}$ (*p-value*). The x-axis represents the physical position of each variant. The red dash line shows the threshold of significance (*p-value* = 0.05). (C) The gene structure of *GRMZM2G064657* and pairwise linkage disequilibriums between markers. The light blue regions represent the upstream and introns. The purple regions represent the exons. (D and E) Comparison of CZ, YA, XSBN, and BLUP between two haplotypes. ***, **, and * denote significant difference between Hap1 and Hap2 at *p-value* < 0.001, *p-value* < 0.01, and *p-value* < 0.05 levels, respectively. "ns" denotes no significant differences (*p-value* > 0.05). (F) Details of two haplotypes. "–" represents a deletion/insertion. N represents the inbred line number of each haplotype.            

and growth stages. Similarly, cigar leaf chlorophyll content also varied greatly during field growth (*Ou et al., 2017*; *Lei et al., 2022*). In maize, the CC of the fifth leaf acts as an accurate predictor to confirm the response of side-dress N fertilizer (*Piekkielek & Fox, 1992*). However, the genetic dissection of chlorophyll content in maize leaves has primarily concentrated on the ear leaves, and there has been a notable lack of research, particularly of the seedling leaf. In this study, we investigated the CC of fifth leaf at seedling stage and ear leaf at filling stage, and revealed that the CC varied extensively in different lines. The heritability estimates of CCFSS and CCEFS were 70.48% and 78.99%, respectively, which was close to that in previous study (*Jin et al., 2023*; *Xiong et al., 2023*), indicating that chlorophyll content is mainly controlled by genotype. Meanwhile, the PVEs of CC-related significant SNPs in this study were mostly less than 10%, which also shows that chlorophyll content is mainly controlled by small-effect polygenes, which further illustrates the
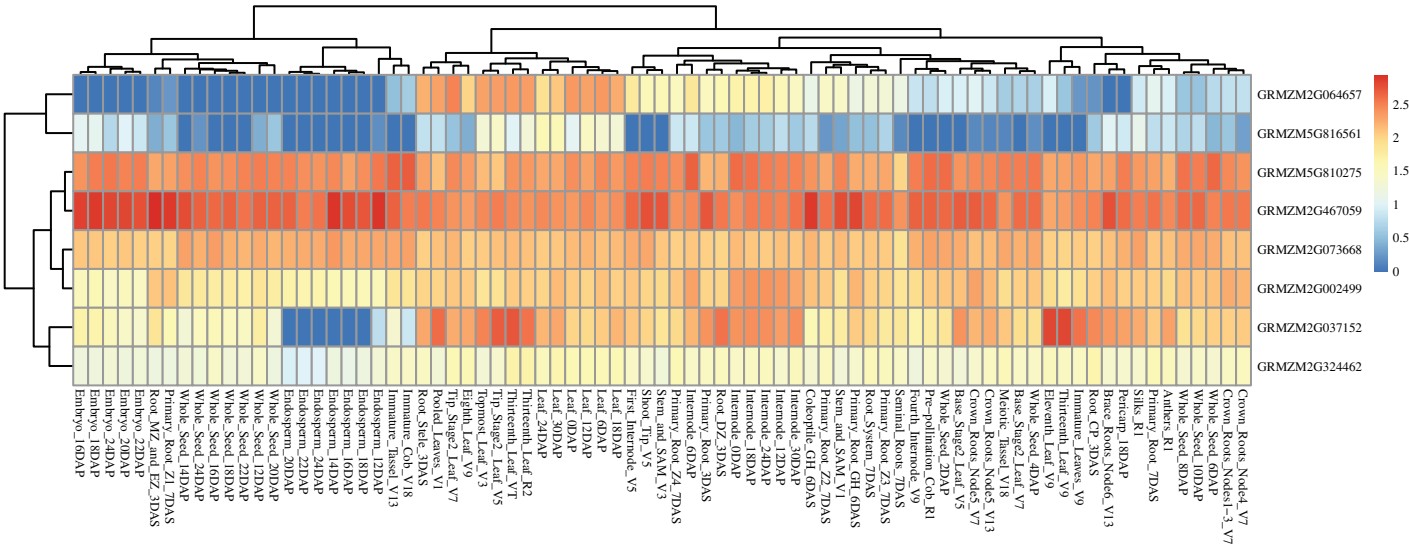

**Figure 7** **Heat map of the expression patterns of eight hub candidate genes.** The value used in the figure is the $\log_{10}$ (Exp +1) conversion ratio of the counts of standardized PRKM in different maize tissues in various development stages. Columns and rows are clustered according to expression similarity. Compared with different periods of a specific gene, blue, yellow, and red colors represent higher, moderate, and lower expression, respectively.

complexity of the regulation of plant chlorophyll content. Remarkably, this study represents the first comprehensive investigation of the genetic foundations of chlorophyll content at both the seedling and mature stages in maize, offering novel insights into the intricate genetic regulation of this essential trait.

As an effective statistical method, GWAS is extensively employed to elucidate the genetic basis of complex quantitative traits and to explore variations and candidate genes related to various agronomic traits (*Guo et al., 2020*; *Yuan et al., 2023*; *Liang et al., 2023*). In this study, 15 CC-related SNPs and 177 candidate genes were identified using the GWAS method. Among these candidate genes, a known gene *GRMZM2G064657* (encoding a phosphate transporter protein) was identified and had been reported to regulate the leaf color (*Li et al., 2021*). This gene was specific expressed in maize leaf, especially in tip stage 2 leaf v7 (https://www.maizegdb.org/gene_center/gene/GRMZM2G064657#rnaseq, accessed 24 Aug 2024). The homologous gene in Arabidopsis, *AT1G68740*, acts at the root level to influence Pi transport and homeostasis, thereby affected the synthesis of chlorophyll and plant growth (*Stefanovic et al., 2007*). The homologous protein OsPHT4 in rice plays a crucial role in the distribution of phosphate ions between the cytoplasm and organelles such as the chloroplast or Golgi apparatus, and it is also implicated in stress responses (*Ruili et al., 2020*). Consistent functionality has also been reported in Chlamydomonas (*Tóth et al., 2024*) and Soybean (*Wei et al., 2023*). Furthermore, *GRMZM2G128644* encodes a VQ motif-containing protein, which belongs to a class of plant specific proteins with a conserved single short FxxhVQxhTG amino acid sequence motif and plays important roles in regulating various developmental processes, such as responding to biotic and abiotic stress, seed development, and photomorphogenesis (*Jing & Lin, 2015*). In a previous study, this gene was identified as a kernel numbers per row

(KNR)-related gene (*Zeng et al., 2022*). Additionally, plants often exhibit a decreased tendency in chlorophyll content under abiotic stresses, which is accompanied by an increase in oxygen free radicals, hydrogen peroxide, and NADH oxidase content (*Chen et al., 2021*). Furthermore, the C2H2 zinc finger protein plays a pivotal role in plant growth and development, as well as in responses to salt, low-temperature, and drought stress. Notably, overexpressing the C2H2-type zinc finger protein gene *RHL41* significantly improves resistance to high-light conditions, evidenced by dramatic changes in plant morphology and increased levels of anthocyanin and chlorophyll (*Asako et al., 2000*). All these findings support the credibility of the CC-related genes identified in this study and significantly augment the existing gene resources pertaining to the synthesis and metabolism of chlorophyll in maize, providing a deeper understanding of the genetic mechanisms of CC.

Researches have shown that chlorophyll content in maize leaves is a crucial determinant of the photosynthesis rate. In this study, 334 maize inbred lines were used to analyze the chlorophyll content in a natural maize population. The findings revealed a high degree of polymorphism in chlorophyll content within this population. In addition, the association panel consisted of six sub population, namely Tropical, PA, PB, Reid, BSSS, and North (*Zhang et al., 2016*), including several elite inbred lines B73, Mo17, Qi319, and so on. These lines were widely used in conventional and molecular breeding programs in southwest of China.

Molecular breeding offers an excellent opportunity to speed up maize improvement programs, especially since a large number of phenotype-related markers and genes have been identified. Currently, marker-assisted selection (MAS) and genome editing (CRISPR-Cas9) have become routine components of maize breeding programs (*Prasanna et al., 2020*; *Xu et al., 2020*; *Hernandes-Lopes et al., 2023*). However, gene discovery remains one of the bottlenecks for the widespread adoption of these technologies in crop breeding (*Scheben & Edwards, 2018*; *Song et al., 2023*). In this study, we identified 15 CC-related SNPs and 177 CC-related candidate genes that can be used to cultivate new maize varieties with higher CC at seedling or filling development stages. Given the importance of higher CC for crop production, we designated the allele with the positive effect as the superior allele. Among 30 elite maize inbred lines widely used in Southwest China, eight lines contained more than five superior alleles (Fig. 4). This finding highlights the potential of these alleles in breeding programs, suggesting that the allelic loci related to high chlorophyll content are closely linked to agronomic traits of interest to breeders and are more likely to be retained during artificial selection and breeding. Taken together, these SNPs and genes can facilitate molecular breeding practice on maize lines with high chlorophyll content and potentially higher yields.

## CONCLUSIONS

In this study, we firstly comprehensively investigated the genetic basis of chlorophyll content in both seedling and ear leaf stages in maize, offering new insights into the complex genetic regulation of chlorophyll. We found that chlorophyll content varied widely among different maize lines, with heritability estimates of 70.84% and 78.99% for

CCFSS and CCEFS, respectively. Subsequently, we constructed a GWAS for these traits across different environments and BLUP based on 43,729 high-quality SNPs. This analysis identified 15 CC-related SNPs and 177 candidate genes. Among these 177 initial candidate genes, eight were consistently identified in at least two environments and were enriched in the top 20 GO terms or KEGG pathways. Further, gene-based association analysis revealed that the upstream region of *GRMZM2G064657* harbored two haplotypes, Hap1 (elite haplotype, AA-) and Hap2 (G-G). Notably, this gene exhibited higher expression levels in leaves compared to other tissues. Consequently, *GRMZM2G064657* was identified as a core regulator affecting chlorophyll content in maize. These findings enhance our understanding of the genetic architecture of chlorophyll content in maize and provide valuable insights for breeding high photosynthetic efficiency varieties.

## ACKNOWLEDGEMENTS
We appreciate the students who assisted with the phenotypic survey.

### Funding
This work was supported by the State Key Laboratory of Crop Gene Exploration and Utilization in Southwest China, SAU (SKL-KF202323), the Natural Science Foundation of Sichuan Province (22NSFSC0148) and the Scientific research initiation project of Mianyang Teachers' College (QD2021A03). The funders had no role in study design, data collection and analysis, decision to publish, or preparation of the manuscript.

### Grant Disclosures
The following grant information was disclosed by the authors:
State Key Laboratory of Crop Gene Exploration and Utilization in Southwest China, SAU: SKL-KF202323.
Natural Science Foundation of Sichuan Province: 22NSFSC0148.
Mianyang Teachers' College: QD2021A03.

### Competing Interests
The authors declare that they have no competing interests.

### Author Contributions
- Peng Liu performed the experiments, analyzed the data, prepared figures and/or tables, authored or reviewed drafts of the article, funding acquisition, and approved the final draft.
- Chenchaoyang Xiang performed the experiments, analyzed the data, prepared figures and/or tables, and approved the final draft.
- Kai Liu performed the experiments, prepared figures and/or tables, and approved the final draft.
- Hong Yu performed the experiments, prepared figures and/or tables, and approved the final draft.

- Zhengqiao Liao performed the experiments, prepared figures and/or tables, funding acquisition, and approved the final draft.
- Yaou Shen conceived and designed the experiments, authored or reviewed drafts of the article, and approved the final draft.
- Lei Liu conceived and designed the experiments, authored or reviewed drafts of the article, and approved the final draft.
- Langlang Ma conceived and designed the experiments, authored or reviewed drafts of the article, and approved the final draft.

## Data Availability

The data is available at figshare: Liu, Peng (2024). Raw data in study of maize chlorophyll content. figshare. Dataset. https://doi.org/10.6084/m9.figshare.26355523.v1.

## Supplemental Information

Supplemental information for this article can be found online at http://dx.doi.org/10.7717/peerj.18278#supplemental-information.

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
