# Peer review of "Genome-wide association study reveals genetic basis and candidate genes for chlorophyll content of leaves in maize (Zea mays L.)"

_PeerJ, doi:10.7717/peerj.18278_

## Round 0.1 · original submission · Major Revisions

Dear Authors

The manuscript cannot be accepted for publication in its current form. It needs a major revision before publication. The authors are invited to revise the paper considering all the suggestions made by the reviewers.

With Thanks

Reviewer 1 ·

Basic reporting

No comment

Experimental design

The study demonstrates an acceptable experimental design.

Validity of the findings

No comments

Additional comments

This research provides valuable insights into the genetic basis of chlorophyll content in maize. The findings may be useful for breeding maize varieties with improved chlorophyll content and potentially higher yield. Chlorophyll content is a physiological trait as it is a result of complex physiological processes within a plant, including photosynthesis, nutrient uptake and utilization, and hormonal regulation. By studying the genes that control CC, breeders can develop maize varieties with improved photosynthetic efficiency and yield.
Comments and Suggestions for Authors
- Abstract
-The abstract is well written.
- Introduction
The introduction could be strengthened by providing more specific examples of how increased chlorophyll content leads to yield improvements.
- The research gap could be more explicitly stated. For example, you could highlight the limited number of identified genes affecting chlorophyll content in maize compared to other crops.
- Materials & Methods
- This section is well-written and provides a good foundation for understanding the study's methodology.
- In line 125, (QTNs) When introducing a term or acronym for the first time in a text, it's essential to write it out fully to provide context for readers. Only after establishing the full term should you use the abbreviation or acronym. Correct to quantitative trait nucleotides (QTNs). Do the same with the whole manuscript.
- Results
- The results section provides a clear and comprehensive overview of the study's findings.
- Consider using more precise or varied vocabulary to enhance the text. For instance, instead of "large variations," you could use "substantial differences" or "significant variability."
- Discussion
- The discussion section effectively summarizes the key findings of the study and places them in the context of existing research.
- Remove the subtitles in discussion section.
- While the discussion mentions a few candidate genes and their potential roles, it could be expanded to provide more in-depth analysis of the identified genes and their functional implications for chlorophyll content. This would strengthen the overall impact of the study.
- The discussion could be strengthened by outlining potential future research directions based on the study's findings. For example, the authors could discuss how their identified SNPs and candidate genes could be used in marker-assisted selection or gene editing to improve chlorophyll content and yield.
- Conclusion
- The conclusion accurately summarizes the study's outcomes; it could be strengthened by briefly discussing the broader implications of the findings. For example; "These results contribute to our understanding of the genetic architecture of chlorophyll content in maize and provide valuable insights for developing breeding strategies to improve photosynthetic efficiency and crop yield."
- Emphasize the novel aspects of your research. For instance: "This study is among the first to comprehensively investigate the genetic basis of chlorophyll content in both seedling and ear leaf stages of maize, providing new insights into the complex genetic regulation of this important trait."

Reviewer 2 ·

Basic reporting

Thank you for considering me to review the manuscript titled “Genome-wide association study reveals genetic basis and candidate genes for chlorophyll content of leaves in maize (Zea mays L.)". The manuscript presents significant findings that contribute to our understanding of the genetic basis of chlorophyll content in maize. However, minor revisions are necessary to ensure clarity, accuracy, and completeness of the reported research. Addressing these points will enhance the manuscript's readability and ensure that the findings are presented robustly and scientifically.
Suggestions:
The manuscript requires a thorough language review to improve clarity and readability.
The abstract needs to be enhanced by including more details on the methodology applied in the study. The abbreviation "GO" on line 31 should be clarified upon its first appearance.
The introduction could be improved by providing more detailed context about the importance of Genome-Wide Association Studies (GWAS) in exploring the genetic basis of important traits. Highlighting how GWAS enables high-resolution mapping and the discovery of multiple loci contributing to trait variation. Improve the explanation of the significance of chlorophyll content in plants, particularly in maize. Use previous studies to support these points and highlight the direct link between high chlorophyll content, increased biomass production, and grain yield. Expand on the knowledge gap this study aims to fill. Provide more background on why this study is significant and how it addresses the limitations of previous research. Use more recent references to support your statements and highlight the novelty and potential impact of your findings. Clearly state the objective of the study and how it aims to contribute to the existing body of knowledge.
Ensure that scientific names (e.g., Zea mays L., in line 73) are italicized throughout the manuscript.
The sections of Ms&Ms and results are well-written and structured. Ensure that raw data is accessible, it appears that the raw data is currently not included in the supplementary files. The resolution of Figures 1-7 should be improved to enhance visual clarity.
Expand the discussion section to include a broader comparison with existing literature, highlighting the novelty and implications of your findings. Provide a more thorough interpretation of the results, discussing the biological significance of the identified SNPs and candidate genes in greater detail. Explain how these findings compare to previous studies and their potential impact on maize breeding programs.
Ensure consistent use of journal abbreviations and proper citation formatting throughout the manuscript. Please review and standardize the reference section according to style guidelines of PeerJ.

Experimental design

The experimental design is appropriate for the study objectives.

Validity of the findings

All data have been provided, ensuring they are robust, statistically sound, and well-controlled. The conclusions are clearly stated, directly linked to the original research question, and appropriately limited to the supporting results.

Reviewer 3 ·

Basic reporting

Good

Experimental design

Needs improvement

Validity of the findings

Resutls can be validated further

Additional comments

Enclsoed

Annotated reviews are not available for download in order to protect the identity of reviewers who chose to remain anonymous.

Reviewer 4 ·

Basic reporting

Abstarct
Summary information is given in the Abstract section.


" It is also necessary to specify whether the corn materials used in the study are early, mid-maturity or late-maturity. It is essential that this statement be reported in detail in the Material-Method section in order to shed light on the studies to be conducted. "

Experimental design

Introduction section
In the introduction section, information is conveyed by quoting 11 publications

Validity of the findings

Materials & Methods
334 corn lines were mentioned in the study.

" What type of corn are these inbred lines? (Dent, flint, pop corn or sweet corn: ) It would be correct to report these. "

It is stated which type of equipment is used to obtain SPAD values. This is a correct step
In the material-method section, how phenotypic data were collected and analyzed is explained.


On the other hand, it is stated how the “Genome-wide association study” was conducted.

Additional comments

Conclusions
In the conclusion section, the importance of the chlorophyll contents of the fifth leaf in the seedling stage and the spike leaf in the filling stage is emphasized.
On the other hand, the connection between molecular selection and Chlorophyll content (CC) has been explained.



I find it appropriate to publish the study after the parts I put in quotes are subject to completion.

Annotated reviews are not available for download in order to protect the identity of reviewers who chose to remain anonymous.

---

## Round 0.2 · accepted · Accept

Dear Authors,

I am pleased to inform you that the manuscript has improved after the last revision and can be accepted for publication.

Congratulations on accepting your manuscript, and thank you for your interest in submitting your work to PeerJ.

With Thanks

Reviewer 1 ·

Basic reporting

no comment

Experimental design

no comment

Validity of the findings

no comment

Additional comments

The authors have made the changes I suggested in the last review. I recommend its publication in this journal.

Reviewer 2 ·

Basic reporting

The authors have effectively addressed all my previous comments and made significant improvements to their manuscript. It is now suitable for acceptance in its current form.

Experimental design

: The experimental design is appropriate for the study objectives

Validity of the findings

All data have been provided, ensuring they are robust, statistically sound, and well-controlled.

Reviewer 3 ·

Basic reporting

Adequate

Experimental design

The methodology applied in this study improved significantly.

Validity of the findings

Authors have made significant improvement in the manuscript as per the suggestions of the reviewers.

Additional comments

Authors have made significant improvement in the manuscript as per the suggestions of the reviewers. The manuscript therefore can be accepted for publication